# CRISPR Editing Enables Consequential Tag-Activated MicroRNA-Mediated Endogene Deactivation

**DOI:** 10.3390/ijms23031082

**Published:** 2022-01-19

**Authors:** Panayiota L. Papasavva, Petros Patsali, Constantinos C. Loucari, Ryo Kurita, Yukio Nakamura, Marina Kleanthous, Carsten W. Lederer

**Affiliations:** 1Department of Molecular Genetics Thalassemia, The Cyprus Institute of Neurology and Genetics, Nicosia 2371, Cyprus; panayiotap@cing.ac.cy (P.L.P.); petrospa@cing.ac.cy (P.P.); loucari.constantinos@gmail.com (C.C.L.); marinakl@cing.ac.cy (M.K.); 2Cyprus School of Molecular Medicine, Nicosia 2371, Cyprus; 3Research and Development Department, Central Blood Institute, Blood Service Headquarters, Japanese Red Cross Society, Koto-ku, Tokyo 135-8521, Japan; r-kurita@jrc.or.jp; 4Cell Engineering Division, RIKEN BioResource Research Center, Tsukuba 305-0074, Japan; yukio.nakamura@riken.jp

**Keywords:** gene tagging, cell/tissue-type-specific gene therapy, γ-globin induction, hemoglobinopathies, BCL11A, erythroid-specific, HUDEP-2 cells, CD34+ cells

## Abstract

Molecular therapies and functional studies greatly benefit from spatial and temporal precision of genetic intervention. We therefore conceived and explored tag-activated microRNA (miRNA)-mediated endogene deactivation (TAMED) as a research tool and potential lineage-specific therapy. For proof of principle, we aimed to deactivate γ-globin repressor *BCL11A* in erythroid cells by tagging the 3′ untranslated region (UTR) of *BCL11A* with miRNA recognition sites (MRSs) for the abundant erythromiR miR-451a. To this end, we employed nucleofection of CRISPR/Cas9 ribonucleoprotein (RNP) particles alongside double- or single-stranded oligodeoxynucleotides for, respectively, non-homologous-end-joining (NHEJ)- or homology-directed-repair (HDR)-mediated MRS insertion. NHEJ-based tagging was imprecise and inefficient (≤6%) and uniformly produced knock-in- and indel-containing MRS tags, whereas HDR-based tagging was more efficient (≤18%), but toxic for longer donors encoding concatenated and thus potentially more efficient MRS tags. Isolation of clones for robust HEK293T cells tagged with a homozygous quadruple MRS resulted in 25% spontaneous reduction in BCL11A and up to 36% reduction after transfection with an miR-451a mimic. Isolation of clones for human umbilical cord blood-derived erythroid progenitor-2 (HUDEP-2) cells tagged with single or double MRS allowed detection of albeit weak γ-globin induction. Our study demonstrates suitability of TAMED for physiologically relevant modulation of gene expression and its unsuitability for therapeutic application in its current form.

## 1. Introduction

Cell-, tissue- or developmental-stage-specific regulation of gene expression is paramount for complex multicellular organisms. Intricate mechanisms acting at multiple levels of the gene regulation process have evolved to determine spatial and temporal gene expression patterns throughout life [1]. Specificity of spatio-temporal effects of genetic engineering are therefore as important as efficiency and safety for meaningful research outcomes and therapy, as attested by numerous studies on the subject. For instance, tissue-specific promoters and other DNA regulatory elements, such as locus control regions, introns, enhancers, silencers and insulators, have been employed to restrict transgene expression to certain tissues in several studies [2,3,4,5]. Similarly, microRNAs (miRNAs), a large class of evolutionarily conserved small non-coding RNAs regulating gene expression mainly at the post-transcriptional level and exhibiting spatial and temporal-patterns of expression [6], have also been harnessed to achieve specificity of expression. Accordingly, the potential of exploiting the endogenous miRNA machinery to achieve detargeting (i.e., cell/tissue- or stage-specific deactivation) of a therapeutic transgene and reduce toxicity was first recognized in 2006 by the Naldini group [7]. By introducing tandem repeats of miRNA recognition sites (MRSs) as tags into the 3′ untranslated region (UTR) of a transgene, its mRNA was targeted for degradation specifically in cells with sufficiently high expression of the cognate miRNA [7,8,9,10], including more recently in preclinical disease models [11,12]. Similar methods were also applied, e.g., in oncolytic virotherapy to regulate virus tropism, in vaccinotherapy to increase safety of live weakened virus vaccines, and in targeted cancer gene therapy to achieve normal-tissue detargeting [13,14,15,16]. Most recently, endogenous miRNA activity has been employed in a number of approaches to confer cell-specificity to clustered regularly interspaced short palindromic repeat (CRISPR)/CRISPR-associated protein (Cas) (CRISPR/Cas) tools, including in miRNA-responsive miR-Cas9 switch systems, Cas-ON switch systems based on miRNA-controlled expression of anti-CRISPR proteins, and miRNA-mediated guide RNA (gRNA)-releasing systems [17,18,19]. 

In analogy to its application for the regulation of transgenes, MRS tagging of UTRs may similarly be employed to engineer detargeting of endogenes. To this end, knock-in approaches for corresponding sequence tags may be based on canonical non-homologous end joining (NHEJ), microhomology-dependent or homology-directed repair (HDR) pathways. However, even for insertion of short tags with double-stranded (ds) or single-stranded (ss) oligodeoxynucleotides (ODNs) to deliver the desired sequences [20,21,22,23], bulk efficiencies are generally low, particularly in primary cells [23,24,25]. Nevertheless, in a recent study the quantification of transgene silencing for a range of mismatched MRS was complemented with analysis of engineered miRNA-mediated endogene silencing for *BRCA1*. Based on lentiviral transduction of the CRISPR/Cas system and on HDR-mediated integration of miR-17 MRSs in the 3′ UTR of *BRCA1*, the study employed high-throughput sequencing after enrichment of edited DNA and cDNA to allow detection of changes in RNA level [26]. 

Here, we set out to develop tag-activated miRNA-mediated endogene deactivation (TAMED) based on efficient hit-and-run editing, in order to achieve engineered endogenous-miRNA-mediated regulation of endogenes, with potential for research and therapeutic application. Employing integration of synthetic MRSs for suitably expressed miRNAs in the 3′ or 5′ UTR of target genes, TAMED would allow cell-, tissue- or stage-specific regulation of gene expression. In this proof-of-principle study, we targeted *BCL11A*, a master γ-globin repressor with multiple functions in several hematopoietic lineages [27,28,29,30,31,32], with the aim of achieving erythroid-specific downregulation of its expression. Reduced BCL11A expression increases γ-globin expression and ameliorates or even cures β-hemoglobinopathies [33,34], by γ-globin compensating defective β-globin expression or neutralizing pathological β-globin variants. Underlining the relevance of *BCL11A*, a targeted, erythroid-specific gene therapy approach for β-hemoglobinopathies, based on disruption of an erythroid-specific enhancer element of *BCL11A*, is currently being tested in clinical trials (NCT03745287 and NCT03432364) with highly encouraging results, pending long-term follow-up data [35,36]. Given its independence from lineage-specific enhancer elements as targets, TAMED would allow modular use of MRSs and knockdown by cognate endogenous miRNAs for any combination of disease modifiers, so that any effect detected here for *BCL11A* would be of both conceptual and, at sufficient efficiency, potential therapeutic significance. To this end, the present study draws on insights from our own recently published transcriptomic analyses [37], in line with other studies and reviews of hematopoietic and erythroid miRNA expression [38,39,40,41,42,43], to shortlist miRNAs likely suitable for TAMED in erythroid cells, and employs CRISPR/Cas9 to insert erythroid-specific miRNA (erythromiR) MRSs into the 3′ UTR of *BCL11A*, based on the canonical-NHEJ and HDR pathways for editing. Our study shows the potential of miRNAs as versatile tools for research and therapy, investigates and compares methods of tagging cells and uncovers difficulties and pitfalls of implementing TAMED.

## 2. Results

### 2.1. Candidate ErythromiRs for TAMED in Adult Late-Erythroid Cells

Employment of TAMED for lineage-specific deactivation relies on exploitation of abundantly and selectively expressed miRNAs. For erythroid-specific repression of *BCL11A*, existing studies suggested several miRNAs with erythroid-specific expression (erythromiRs) as suitable candidates for TAMED [38,39,40]. Among them, miR-451a is characterized by abundant late-erythroid expression and low expression in HSCs and non-erythroid lineages [39,44,45,46,47,48], and according to miRDB is not predicted to naturally target *BCL11A* [49]. The high expression of miR-451a was recently confirmed in our transcriptomic analysis of erythroid differentiation cultures of adult-type primary CD34+ and human umbilical cord-blood-derived erythroid progenitor-2 (HUDEP-2) cells (NCBI GEO accession ID GSE165011A) [37], which established miR-451a for both cell models as the most highly expressed miRNA in all late-stage erythroid cells and as significantly upregulated during erythroid differentiation (mean log_2_ fold change of 4.39). The same dataset was utilized to shortlist 73 additional differentially expressed erythromiRs in late- vs. early-erythroid samples based on their high and highly upregulated expression in adult late-erythroid cells, which would thus be suitable for alternative or multiplexed application of TAMED to target *BCL11A* or other erythroid disease modifiers (see Figure 1 for the top 20 differentially expressed miRNAs and Appendix A for all 73 miRNAs).

### 2.2. Design Choices for TAMED of BCL11A

We then selected the BCL11A-XL isoform as the specific target transcript for TAMED, because it is particularly abundant in erythroid cells [50] and thus of greatest therapeutic relevance. Both dsODNs (for integration via the c-NHEJ pathway) and ssODNs (for integration via the HDR pathway) were designed to carry 1–4 fully miR-451a-complementary MRSs, for integration into the 5′ and 3′ UTRs of BCLL1A-XL in the highly robust human embryonic kidney 293T (HEK293T) cells, in erythroid HUDEP-2 cells and in primary CD34+ cells. In silico analyses of sequence motifs, secondary structures and known MRSs allowed exclusion of UTR sequences potentially critical to gene expression. Thus, a total of seven likely inert BCL11A-XL UTR sites, i.e., sites where indel formation alone, without the insertion of MRSs, would not affect gene expression, were chosen for initial experimental evaluation. An overview of the seven UTR sites (three in the 5′ UTR and four in the 3′ UTR) and the oligos and corresponding modes of delivery used in the study is shown in Figure 2.

### 2.3. Validating Cleavage and Inertness of Shortlisted UTR Target Sites

To test inertness of UTR target sites shortlisted for potential tag integration, gRNAs were designed for all seven BCL11A-XL UTR sites to allow targeted DSB induction and functional evaluation of the resulting disruption events. gRNAs were individually cloned into lentiCRISPRv2 plasmid vector and tested for their cleavage efficiency by expression from integrating lentiviral vectors (LVs) in HUDEP-2 cells. A mock-transduced sample (MOCK) and a Cas9-only-transduced sample (EMPTY) were included in experiments as negative controls, whereas a cell sample transduced with a gRNA targeting the start codon of *BCL11A* (SC) was included as a positive control for γ-globin induction. CRISPR/Cas9-mediated genome editing efficiency at the target loci, as evaluated by T7 endonuclease I assay (T7EI) assay, varied in the range of ~3% to 52% (Figure 3a,b). Four gRNAs (5′ UTR 1, 5′ UTR 3, 3′ UTR 1 and 3′ UTR 4) showing the highest genome editing efficiency were further analyzed for their effect on BCL11A and γ-globin expression. BCL11A immunoblot analysis confirmed the neutral effect of editing on gene expression for at least two gRNAs, 3′ UTR 1 and 5′ UTR 1 (Figure 3c), while reversed-phase high-performance liquid chromatography (RP-HPLC) analysis of γ-globin expression in HUDEP-2 cells after erythroid differentiation showed absence of γ-globin expression for all gRNAs but the SC control (Figure 3d). Taken together, the results of gene editing analysis showed that CRISPR/Cas9-mediated cleavage/modification of 5′ and 3′ UTR could be achieved at a subset of selected sites without interfering with BCL11A expression and function.

Based on these results and to optimize conditions for genomic integration of donor DNA, the 3′ UTR 1 gRNA was selected. In addition to meeting other quality criteria by in silico analyses, the 3′ UTR 1 site 593 nt upstream the 3′ end of the BCL11A-XL 3206-nt 3′ UTR (Figure 2) was selected as potentially advantageous, based on active MRSs mostly being located near the 3′ end of 3′ UTRs of target mRNAs instead of more centrally [52,53]. 

### 2.4. NHEJ-Mediated Integration of MRSs of miR-451a as dsODNs

For NHEJ-mediated integration, blunt-ended dsODNs (Appendix A) were designed to introduce MRSs into DSB sites, in analogy to designs initially employed for GUIDE-Seq technology [22] and in preference to ssODNs, which reportedly show inferior performance for NHEJ-mediated insertion [54]. Two phosphorothioate bonds were incorporated on both the 5′ and 3′ end of dsODNs to confer nuclease resistance [55]. A 5′ phosphate group was also included in the sequence to facilitate ligation reactions [56]. dsODNs contained tandem repeats of either two or four direct (head-to-tail) MRSs with perfect complementarity to miR-451a in order to allow cleavage of mRNA and high suppression of the target gene with minimal sponge effect [57]. A HaeIII restriction site was included in dsODN templates to facilitate restriction fragment length polymorphism analysis of PCR-amplified fragment (PCR-RFLP)-based detection of integration (Figure 2). dsODNs and CRISPR/Cas9 (as RNPs) were either co-delivered or sequentially delivered in HEK293T, HUDEP-2 and primary CD34+ cells by nucleofection. A mock-transfected sample (MOCK) and an RNP-only-transfected sample (NO DONOR) were included in all experiments as negative controls.

In HEK293T cells, dsODNs bearing two or four direct MRSs for miR-451a (dsODN451-2MRSs or dsODN451-4MRSs) at 5 and 20 pmole and CRISPR/Cas9 RNPs were co-delivered by nucleofection. Assessment of genome editing by T7EI assay showed efficiencies from 32.6% to 44.8% (Figure 4a,b), while no cytotoxicity was observed in cell cultures. PCR-RFLP analysis using the HaeIII enzyme failed to detect amplicon cleavage as proxy for donor DNA integration (Figure 4c). To detect if low frequency integration events (below the detection limit of PCR-RFLP) were present in our cells, we designed a set of PCR primers for dsODN-specific amplification, one forward primer (DONOR 451 FW), binding inside the donor DNA sequence, and one reverse primer (3′ UTR 1 RV) binding in the downstream 3′ UTR-encoding genomic DNA (gDNA) sequence. PCR products of approximately 250 and 302 bp were indicative of correctly oriented genomic integration events of two and four MRSs, respectively (Figure 4d). Importantly, in the original publication of the GUIDE-Seq method, 5 pmole of similarly modified dsODN (GUIDESeq-dsODN, 34-bp long, Appendix A) were efficiently integrated in HEK293T cells by co-nucleofection of plasmid-encoded nucleases [22]. Employing the identical GUIDE-Seq dsODN as positive control for our experimental setup, we achieved up to 45% integration efficiency (Appendix A), vindicating delivery choices and chemistries used here, but indicating sequence-dependent efficiency as a major drawback of NHEJ-based tagging with novel MRSs.

In HUDEP-2 cells as the immortalized equivalent of primary erythroid cells, dsODNs bearing two direct MRSs for miR-451a (dsODN451-2MRSs) at 50, 100, 150, 200 pmole and CRISPR/Cas9 RNPs were co-delivered by nucleofection. Both T7EI assay and Sanger-sequencing-based deconvolution analysis by Tracking of Indels by Decomposition (TIDE) were performed for the assessment of targeted genome editing efficiency of RNPs at the target locus (3′ UTR 1), six days after nucleofection (Figure 5a–c). The percentage of genome editing was high (~67% and 77.4% according to T7EI assay and TIDE, respectively) in samples nucleofected only with RNPs (NO DONOR), while a decline in the percentage of editing was observed with increasing amount of donor. Delivery of dsODNs induced marked, dose-dependent cytotoxicity in HUDEP-2 cells (Figure 5d), also in agreement with previous publications for phosphorothioated ODNs [58,59]. NHEJ-mediated capture of dsODNs into CRISPR/Cas9-mediated DSBs as assessed by PCR-RFLP using HaeIII enzyme was once more below the method’s detection limit (data not shown). However, dsODN-specific amplification detected the presence of integration events in HUDEP-2 cells, as well as in primary CD34+ cells (with similar toxicity), even when lower donor quantities (5 and 20 pmole) were delivered to cells (Figure 5e). Replication with sequential delivery of tools (see Section 4.5) in HUDEP-2 and CD34+ cells showed no improvement in either toxicity or integration efficiency (data not shown). Of note, delivery of longer dsODNs carrying four MRSs in HUDEP-2 cells induced marked toxicity precluding any functional analyses in corresponding cells.

### 2.5. HDR-Mediated Integration of MRSs of miR-451a as ssODNs

For HDR-mediated integration of MRS tags, 100 pmole of ssODNs bearing one MRS (ssODN451TS-1MRS, ssODN451NTS-1MRS) or two direct-repeat MRSs (ssODN451TS-2MRSs, ssODN451NTS-2MRSs and Alt-R HDR-2MRSs) for miR-451a, and CRISPR/Cas9 RNPs were co-delivered in HUDEP-2 cells by nucleofection. ssODNs with more than two MRSs were not designed, as publications on HDR-based capture of ssODNs at the time showed that optimal donor design (balancing HDR knock-in efficiency and cytotoxicity) entails 30–35 nt homology arms and a total length of no more than 100 nt [60,61,62,63,64]. In all HDR experiments, a mock-transfected sample (MOCK) and an RNP-only-transfected sample (NO DONOR) were included as negative controls. To enhance HDR efficiency, which is inherently low in several cell types, including HUDEP cells [65], we used (a) Alt-R HDR enhancer (a small molecule compound that inhibits the NHEJ pathway and thus shifts the balance in favor of the HDR pathway [66,67]) and (b) nocodazole (a cell cycle synchronization agent that synchronizes cells at G2/M phase where the HDR pathway is more active [68,69]). PCR-RFLP analysis, performed ~72 h after nucleofection, showed HDR-mediated integration of MRSs in the 3′ UTR of BCL11A-XL in the range of 5% to 32%, with marginal increases by Alt-R HDR enhancer, but not by nocodazole (Figure 6). At every instant, delivery of the longer ssODN451TS-2MRSs/ssODN451NTS-2MRSs donors was toxic to HUDEP-2 cells, and corresponding cultures did not survive to allow functional analyses. A pool of HUDEP-2 cells with 18% integration of the shorter ssODN451TS-1MRS survived to be further analyzed.

### 2.6. Characterization of Monoclonal Cell Populations of HEK293T and HUDEP-2 Cells Bearing MRSs for miR-451a

To facilitate downstream phenotypic characterization, clones bearing MRS tags for miR-451a were isolated from heterogeneous polyclonal HEK293T and HUDEP-2 cell populations after editing, for expansion and analysis as single samples (n = 1). For bulk HEK293T cell populations treated with donors carrying two or four MRSs, limiting dilution in two 96-well plates allowed isolation of 66 single-cell colonies that survived the initial expansion phase, of which four were positive clones that had MRSs in the correct orientation in the 3′ UTR (efficiency ~6%) (Figure 7a). For bulk HUDEP-2 cell populations treated with dsODNs carrying two MRSs, sib selection was applied as a two-phase isolation method for rare mutants, which allowed isolation of four first-round positive oligoclonal cell populations from a 96-well plate, before identification of nine positive clones after second-round selection. Finally, cloning by limiting dilution of pools of HUDEP-2 cells treated with ssODNs resulted in the isolation of only one clone carrying one MRS for miR-451a. Therefore, clonal isolation allowed comparison with unedited cells at the DNA and protein level and assessment of genotype-phenotype relationships for a total of four HEK293T clones and ten HUDEP-2 clones bearing one or more correctly oriented MRSs for miR451. For genotyping, Sanger DNA sequencing of the PCR-amplified targeted genomic locus proved to be challenging, owing to long mononucleotide repeats, common for UTR regions [70,71], near the target site, causing DNA polymerase replication slippage and stutter products (Figure 7b). Sequencing of clones was thus based on design of internal, DSB-proximal sequencing primers, manual deconvolution of sequencing data and combination of chromatograms for both strands [72]. For NHEJ-mediated integration of dsODNs, imprecise sequence insertions, i.e., donor sequence insertions with loss of flanking target or donor sequences resulting in combined knock-ins and indels, were detected in all instances (Figure 7c). 

At the protein level, tagging of HEK293T cells with MRSs for miR451a resulted in a reduction in BCL11A expression by up to 25% in a homozygous clone carrying four MRSs for miR-451a, and up to 36% after transfection of hsa-miR-451a miRNA mimic in the same clone (Appendix A). Observations for the miRNA mimic suggest that at sufficiently high miR-451a expression, MRSs would have resulted in substantial BCL11A reduction for four MRSs in HEK293T cells. Analysis of HUDEP-2 clones carrying at most two MRSs revealed increased γ-globin expression after erythroid differentiation compared with controls, 46.8-fold by immunoblot analysis, and up to 8.3-fold by RP-HPLC, with results showing high concordance between the two methods of analysis as calculated by Pearson Correlation Coefficient analysis (r = 0.952, R^2^ = 0.907, *p* < 0.0001) (Appendix A). Conversely, immunoblot analysis of HUDEP-2 clones for BCL11A expression on day 4 of erythroid differentiation (intermediate stage) showed variable results for BCL11A expression, indicative of a greater effect of sampling and differentiation parameters for the highly developmentally regulated BCL11A and miR-451a than simplex or duplex MRS tags (Appendix A). In summary, results at the protein level for single samples (n = 1) per clone demonstrated proof of principle of TAMED, although the achieved induction of γ-globin was small and below the therapeutic cut-off level for β-hemoglobinopathies.

## 3. Discussion

This study explores TAMED, a method that utilizes CRISPR/Cas9-based DSB induction and NHEJ- or HDR-mediated incorporation of MRS tags in the 3′ UTR of an endogene to turn the endogene mRNA into a target for degradation by the cognate miRNA.

Harnessing endogenous miRNAs in synthetic miRNA-regulated systems for both research and therapy has been extensively studied [73]. However, harnessing endogenous miRNAs to control expression of endogenes remains largely unexplored. Here, CRISPR/Cas-based NHEJ and HDR repair mechanisms were employed for integration of synthetic MRSs for the erythromiR miR-451a in the 3′ UTR of BCL11A-XL, aiming to achieve erythroid-specific downregulation of its expression and therefore induction of γ-globin expression. Conceptually, other γ-globin repressors could be similarly targeted in simplex and multiplex applications, to modify disease severity by their lineage-specific suppression. Disruption of the *BCL11A* erythroid-specific enhancer for erythroid-specific knockdown of *BCL11A*, which is currently being tested in clinical trials, depends on the existence and discovery of gene regulatory elements conferring tissue specificity of gene expression. By contrast, TAMED can be applied more broadly as a therapeutic or research approach also to largely uncharacterized target genes for their spatial or temporal repression, as long as corresponding miRNA expression profiles are known. Moreover, TAMED allows precise regulation of the magnitude of gene suppression (by modulating, for example, the number of MRSs or the extent of miRNA-MRS complementarity), rather than the “all-or-none” knockout of CRISPR/Cas9-mediated-effect, similar to the result obtained with shRNA technology but without the corresponding permanent toxicity [74,75,76]. TAMED may thus also facilitate lineage-specific functional analyses of endogenes in general, and provide tools and components that can be used in a modular fashion and in a range of applications.

In this study, either dsODNs (for integration via the NHEJ pathway) or ssODNs (for integration via the HDR pathway) were delivered along with RNPs targeting the 3′ UTR 1 site of BCL11A-XL in HEK293T and HUDEP-2 cells. Initial analysis of random on-target NHEJ-mediated indel formation on gene expression showed that CRISPR/Cas9-mediated cleavage/modification of 5′ and 3′ UTR can be achieved at selected sites without interfering with gene expression. The delivery of phosphorothioate-modified dsODNs and ssODNSs induced substantial toxicity in HUDEP-2 but not in HEK293T cells, which were correspondingly more amenable to transfection with longer donors, such as dsODNs carrying four MRSs. After optimizing nucleofection conditions and donor concentration to achieve the best ratio for integration efficiency to cytotoxicity, we reached a maximum of 6% correctly oriented integration of dsODN451-4MRSs by NHEJ in HEK293T cells, and 18% of ssODN451TS-1MRS by HDR in HUDEP-2 cells. Although higher integration rates (up to 32%) of Alt-R HDR-2MRSs by HDR were also achieved in HUDEP-2 cells, expansion and differentiation of cells for γ-globin expression analysis were impaired by high cytotoxicity. Nocodazole, a cell cycle synchronizer that has been shown to promote HDR in other cell lines [69,77,78] was applied in HUDEP-2 cells here for the first time, but failed to increase HDR rates. Of note, we reproduced high NHEJ-based integration efficiencies in HEK293T cells for the dsODN used in the originally published GUIDE-Seq method, showing identical oligo chemistry but different sequence composition and length to our donors (34 nt vs. 48 nt) [22]. This suggested a significant and unpredictable effect of the specific dsODN sequence and length on integration efficiencies, indicative of practical difficulties in any de novo tag design for NHEJ-based TAMED.

Based on our data, utility of TAMED is thus currently limited by two independent effects. As the first limiting effect, NHEJ-mediated integration of dsODNs by CRISPR/Cas9 is nondirectional and imprecise. Nondirectionality for Cas9 brings about that approximately half of all integration events may be reversed and therefore is unproductive for TAMED. Imprecision leads to random on-target indel events of flanking gDNA and of sequence tags, in line with previous studies using CRISPR/Cas9 in knock-in experiments [20,79]. Such ubiquity of indels at CRISPR/Cas9 on-target sites may be caused by blunt-ended DSBs. By contrast, the typically staggered cuts of zinc-finger nucleases produce complementary acceptor ends, which facilitate precise donor sequence insertion of dsODNs with microhomology arms in knock-in experiments [80]. This phenomenon effectively reduces the availability of MRSs for interaction with miRNAs by their partial deletion in truncated insert sequences and might additionally interfere with the functionality of the UTR by extended deletion of DSB-flanking genomic DNA. Both orientation and border precision may be addressed by employing Cas12a (formerly called Cpf1 (CRISPR from Prevotella and Francisella 1)), as in contrast to Cas9-delivered blunt ends, it produces predictable 5′-overhang staggered DSBs, which might increase efficiency of donor DNA integration, also allowing its directional insertion [81]. As the second limiting effect, phosphorothioate-modified dsODNs show high levels of toxicity, exacerbated from HEK293T to HUDEP-2 and primary cells, and from shorter to longer donors. Recently published optimized protocols of the GUIDE-Seq method indicate 3′-only end-protected dsODN tags as less toxic than double-5′-3′-end protected ones, despite that high toxicity in hematopoietic stem cells was communicated as the major limitation of the GUIDE-Seq method [59]. Alternatively, HDR-based tag insertion is precise, inherently directional and overall more efficient than NHEJ-mediated integration, but pronounced toxicity of longer ssODNs limits the ability to provide repeated or multiplexed MRSs for higher efficiency of detargeting. It is tantalizing to speculate that concatenated MRS tags in the 3′ UTR of BCL11A-XL in HUDEP-2 cells may have resulted in a greater increase in γ-globin expression after erythroid differentiation. While the current study was based on cutting-edge ss donor technology provided by one of the leading manufacturers [82], the toxicity may be addressed by future development of advanced donor DNA chemistries. As an alternative solution, the insertion of tags by prime editing technology and an MRS-encoding pegRNA is also conceivable in the future, but delivery for prime editing has not yet been demonstrated for inserts above 44 nt and may face additional challenges in primary cells [83]. On whichever technology TAMED is based, comprehensive analyses of off-target and recombination events will enhance any genotype-phenotype correlation. In particular for therapeutic application, safety and fidelity analyses need to go beyond on-target analyses and off-target predictions as performed here, and would mandatorily include sensitive detection of recombination and off-target events.

Our results in MRS-tagged clones, though based on single samples (n = 1) per clone, strongly support the concept of miRNA-mediated detargeting of endogene expression and gave rise to changes in BCL11A expression with resulting marked changes in γ-globin expression in multiple clones. However, in bulk populations the observed effect on BCL11A and γ-globin expression was small, revealing inefficiency of TAMED in its current form as a therapeutic tool for β-hemoglobinopathies. Moreover, evaluation of miR-451a-mediated TAMED in other lineages is essential for its meaningful application in hematopoietic stem and progenitor cells, as miR-451a also has some expression and role in non-erythroid-lineage cells, such as T cells [45,48]. This calls for a systematic evaluation of alternative miRNAs with suitable expression patterns in erythroid cells (Figure 1, Appendix A). It is possible that other miRNAs with lower expression would be better candidates for erythroid-specific application of TAMED, as it is known that miRNA expression levels do not always correlate with the extent of their knockdown effect on target genes [84]. Several other factors including accessibility of the target region, optimal spacing of MRSs and interaction with other miRNAs variably affect the suppression of target genes [57,85,86,87,88,89]. It is therefore also possible that insertion of MRSs in another region of the 3′ UTR or in the 5′ UTR of *BCL11A* would have resulted in more efficient suppression of BCL11A. Another issue of import for any selected miRNA is whether its isolated exploitation for TAMED will be able to evoke clinically relevant suppression of a target gene given the “multiple-to-multiple” nature of small RNA interactions with targets, in accordance with our own transcriptomic data [37]. Importantly, for the highly expressed miR-451a, our indicative clonal TAMED data show consequential suppression of BCL11A with ensuing γ-globin increases, and as a natural genetic phenomenon, 3′ UTR mutations that abrogate natural or create illegitimate MRSs elsewhere have been shown to cause disease [90,91]. However, these might be exceptional observations, as suggested by several studies and because knockouts of individual miRNAs in erythroid cells have so far failed to produce significant phenotypic changes related to γ-globin [10,92,93,94]. Therefore, insertion of different concatenated MRSs for two or three different erythromiRs may achieve higher gene suppression than achieved employing multiple MRSs for mi-R451a alone, even if the corresponding miRNAs are expressed at moderate levels [95]. This strategy would at the same time reduce the risk of saturating the function of any one cognate miRNA [92].

Finally, in addition to the need to enhance the efficacy of miRNA-mediated suppression, application of TAMED needs to consider miRNA off-target effects as an inherent concern in any miRNA-based therapies and research applications. For instance, it is possible that the introduction of artificial recognition sites will evoke miRNA dysregulation by saturating the endogenous miRNA and by interfering with its ability to regulate its natural targets (sponge effect) [96]. To prevent such saturation in our study, we designed MRSs to be perfectly complementary to the corresponding miRNA, as previous studies have shown that perfect targets accelerate miRNA turnover and decrease the risk of saturating the miRNA [10,97]. Moreover, it is possible that the introduction of foreign genetic material in UTRs will create new or perturb existing unrelated MRSs, which may be addressed by functional analyses and transcriptomic studies once effective combinations of MRSs and integration sites have been established. 

## 4. Materials and Methods

### 4.1. Culture of Human Primary Cells and Cell Lines

CD34+ cells were isolated from peripheral blood of healthy individuals after mononuclear cell isolation using a density gradient medium, in line with procedures originally published as Protocol C [98] and modified as previously described by us [37].

HUDEP-2 cells were expanded in medium based on StemSpan SFEM II (Stem Cell Technologies, Vancouver, BC, Canada) supplemented with 1 µM Dexamethasone (Sigma-Aldrich, Munich, Germany), 5 μg/mL Doxycycline (Clontech Laboratories, Mountain View, CA, USA), 100 ng/mL Recombinant Human Stem Cell Factor (hSCF) (PeproTech, Rocky Hill, CT, USA), 3 IU/mL Epoetin alpha (Binocrit 4000 IU/0.4 mL, Sandoz GmbH, Kundl, Austria), and 2× Penicillin-Streptomycin (Thermo Fisher Scientific, Waltham, MA, USA), at concentrations below 0.5 × 10^6^ cells/mL, following procedures for expansion and erythroid differentiation in a three-phase erythroid differentiation culture system as previously described [37].

HEK293T cells, Takara Bio, Saint-Germain-en-Laye, France, #632180) were maintained in Iscove’s Modified Dulbecco’s Medium (IMDM) (Sigma-Aldrich, Munich, Germany), supplemented with 10% Fetal Bovine Serum Qualified, HI, Standard (Thermo Fisher Scientific, Waltham, MA, USA), 1% GlutaMAX Supplement (Thermo Fisher Scientific, Waltham, MA, USA) and 1× Penicillin-Streptomycin (Thermo Fisher Scientific, Waltham, MA, USA). Cells were passaged to a ratio of 1:10 when reaching 80–90% confluency, using Trypsin-EDTA (Invitrogen, Carlsbad, CA, USA) to detach cells.

### 4.2. gRNA and ODN Design

For the design of gRNA sequences, we used the Zhang Lab CRISPR guide design tool [99]. UTR sites targeted by gRNAs were verified for the absence of (i) functional/regulatory motifs or conserved sequences using UTRdb and RegRNA 2.0 [47,48], (ii) complex secondary structures that could hinder miRNA accessibility to the region by calculating the free energy (ΔG) of the 3′ and 5′ flanking 70-nucleotide (nt) region around the site using Mfold [89,100] and (iii) other validated MRSs by using miRTarBase [101]. For NHEJ-mediated integration of donor DNA, 5′-phosphorylated dsODNs bearing two or four direct MRSs for miR-451a and two phosphorothioate bonds on both the 5′ and 3′ template ends were designed and ordered either as duplex oligonucleotides (oligos) from Metabion, Munich, Germany or as ss oligos from Integrated DNA Technologies (IDT), Coralville, IA, USA. Both duplex and ss oligos were reannealed/annealed as described in 2.7 to form duplexes (48-base pair (bp) dsODN451-2MRSs or 100 bp dsODN451-4MRSs; Appendix A) before delivery to cells. A published 34-bp GUIDESeq-dsODN was ordered as ss oligos from IDT, Coralville, IA, USA to test our experimental setup (Appendix A) [22]. For HDR-mediated integration, ssODNs (Ultramer DNA Oligos) bearing one (92-nt long) or two (118-nt long) direct-repeat MRSs for miR-451a, flanked by 35-nt homology arms surrounding the double-strand break (DSB) site and two phosphorothioate bonds on both the 5′ and 3′ template ends, were designed and ordered from IDT, Coralville, IA, USA. Ultramer DNA oligos were designed to have homology arms identical to either the non-target strand containing the PAM sequence (oligos denoted as “NTS”) or the target strand (oligos denoted as “TS”) for donors with one MRS (ssODN451TS-1MRS, ssODN451NTS-1MRS) and two direct-repeat MRSs (ssODN451TS_2MRSs, ssODN451NTS_2MRSs), to allow efficiency and toxicity evaluation of different donor designs (Appendix A). In selected HDR-based experiments, we additionally used an Alt-R HDR Donor Oligo designed to be complementary to the non-target strand and bear two direct-repeat MRSs for miR-451a (Alt-R HDR-2MRSs, 118-nt long, Appendix A) [82].

### 4.3. Lentiviral Vector (LV) Construction and Production

gRNAs sequences were generated as oligos from Metabion, Munich, Germany (Appendix A) and cloned into lentiCRISPRv2 plasmid vector (Plasmid #52961, Addgene, Watertown, MA, USA) [102]. Briefly, oligo strands (sense and antisense) bearing overhangs complementary to the vector were annealed in a Veriti thermal cycler (Thermo Fisher Scientific, Waltham, MA, USA) under the following conditions: 95 °C for 4 min, 80 °C for 4 min (ramp rate of 10%), 30 °C for 4 min (ramp rate of 0.3%) and pause at 15 °C. The lentiCRISPRv2 backbone was isolated from the plasmid vector by digestion with BsmBI (New England Biolabs, Ipswich, MA, USA), gel-excised and gel-extracted with NucleoSpin^®^ Gel and PCR Clean-Up Kit (Macherey-Nagel, Düren, Germany) according to the manufacturer’s instructions. Rapid DNA Ligation kit (Roche, Basel, Switzerland) with 1:6 vector:insert (*v*/*v*) ratio was applied to ligate annealed oligos into the lentiCRISPRv2 backbone. Bacterial transformation of TOP10 Chemically Competent *E. coli* (Thermo Fisher Scientific, Waltham, MA, USA) was performed according to the manufacturer’s instructions, and colony screening for ligation events of interest was performed by Sanger DNA sequencing using the lentiCRISPRv2-forward primer (Appendix A). An amount of 20 μg of vector was co-transfected with 5 μg of envelope plasmid pMD2.G (encoding VSV-G) (Plasmid #12259, Addgene, Watertown, MA, USA) and 15 μg of psPAX2 (packaging plasmid encoding gag, pol, tat/rev genes) (Plasmid #12260, Addgene, Watertown, MA, USA), using polyethylenimine (linear, 25 kDa, Polysciences Inc., Warrington, PA, USA) into HEK293T cells as described elsewhere [103]. Vector-containing supernatant was harvested at 24, 48 and 72 h post-transfection, filtered through 0.45 μm PVDF filters (Merck Millipore, Burlington, MA, USA) and centrifuged at 20,000× *g* for 4 h at 4 °C. LV pellets were suspended in 100 μL StemSpan SFEM II (StemCell Technologies, Vancouver, BC, Canada) and used directly for cell transduction or stored at −80 °C until use.

### 4.4. Lentiviral Transduction of HUDEP-2 Cells

An amount of 1 × 10^6^ cells in 0.5 mL culture medium, supplemented with 8 μg/mL Polybrene Infection/Transfection Reagent (Merck Millipore, Burlington, MA, USA), was transduced with 50 μL concentrated LV. The cell-virus mixture was incubated at 37 °C, 5% CO_2_ humidified atmosphere for 6 h, during which time it was mixed by hourly gentle pipetting. Then, cells were seeded in fresh culture medium for expansion. Twenty-four hours after viral transduction, puromycin dihydrochloride (Santa Cruz Biotechnologies, Dallas, TX, USA) was added to cultures at 1 μg/mL for positive antibiotic selection of transduced cells for typically 2–4 days.

### 4.5. Nucleofection of Cells

Nucleofection of purified Cas9 protein (PNA Bio, Newbury Park, CA, USA) and synthetic gRNAs (Synthego, Menlo Park, CA, USA) (Appendix A) as ribonucleoprotein (RNP) particles was performed by the 4D-Nucleofector (Lonza, Basel, Switzerland) using the P3 Primary Cell 4D-Nucleofector X Kit (Lonza, Basel, Switzerland) and the CA-137 program for HUDEP-2 and CD34+ cells, and the SF Cell Line 4D-Nucleofector X Kit L (Lonza, Basel, Switzerland) and the CM-130 program for HEK293T cells. Nucleofection was performed as detailed elsewhere [104]. dsODNs were prepared by reannealing/annealing oligos synthesized by Metabion, Munich, Germany or IDT, Coralville, IA, USA (Appendix A). For reannealing/annealing, duplex oligos or equimolar mixtures of sense and antisense strands were heated at 95 °C for 1 min to remove secondary structures and denature partially annealed oligos, and then cooled slowly to 4 °C at a ramp rate of 0.5% to allow for stringent binding of complementary sequences. Both co-delivery and sequential delivery of RNPs and dsODNs were tested [105]. For co-delivery of RNPs and dsODNs, dsODNs were added to the RNP/cell mixture immediately before nucleofection, whereas for their sequential delivery, cells were first nucleofected with RNPs, washed with Dulbecco’s Phosphate-Buffered Saline (DPBS) (without CaCl_2_/MgCl_2_) (Sigma-Aldrich, Munich, Germany), and nucleofected again with dsODNs. ssODNs (Ultramer DNA Oligos or Alt-R HDR Donor Oligos) synthesized by IDT, Coralville, IA, USA (Appendix A) were co-delivered with RNPs in HUDEP-2 cells by nucleofection. In HDR experiments, we used the Alt-R-S.p. Cas9 nuclease V3 (IDT, Coralville, IA, USA) and synthetic gRNAs (Synthego, Menlo Park, CA, USA) at 1:1.2 molar ratio. Alt-R Cas9 Electroporation Enhancer (IDT, Coralville, IA, USA) was added to the cell/RNP/HDR donor mixture just prior to nucleofection. Alt-R HDR Enhancer (IDT, Coralville, IA, USA) was added to the culture medium for 12 h after nucleofection according to the manufacturer’s instructions. HEK293T cells were nucleofected with 0.5–100 pmole of mirVana hsa-miR-451a miRNA mimic (Thermo Fisher Scientific, Waltham, MA, USA) and 24 h after nucleofection analyzed for protein (BCL11A) expression by immunoblotting.

### 4.6. Generation of Clonal Populations of HEK293T and HUDEP-2 Cells

Two days after nucleofection, heterogeneous pools of edited HEK293T and HUDEP-2 cells were quantitated with a hemocytometer and diluted through serial dilutions to a final density of 5 cells/mL in 10 mL fresh expansion medium. For routine clonal selection, 100 μL of this solution was then transferred into each well of a 96-well plate, giving an average seeding density of 0.5 cells/well. Cells were incubated for 5 days undisturbed, and in the case of HUDEP-2 cells, supplemented with additional doxycycline every other day. Then plates were scanned under an inverted light microscope, and wells containing only one colony were expanded for ~20 days, with careful weekly addition or replenishment of media, and addition of doxycycline on alternate days for HUDEP-2 cells. When cell populations in wells reached confluency, cells were transferred to 48-well plates for further expansion. For clonal selection of rare donor DNA integration events, we applied sib selection instead [106]. Briefly, edited cells were plated at a density of 20 cells/well into a 96-well plate and expanded for ~20 days. Oligoclonal populations with detectable integration of donor DNA were then subjected to a second round of selection/cloning, this time plated at a density of 0.5 cells/well into a 96-well plate, as described above. One sample per clone (n = 1) was analyzed.

### 4.7. DNA Analysis

Genomic DNA was extracted from cell pools using QIAmp DNA Blood Mini Kit (Qiagen, Hilden, Germany) and from monoclonal cell populations using Quick Extract™ DNA Extraction Solution (Lucigen, Middleton, WI, USA), according to the manufacturer’s instructions. Plasmid DNA from bacterial mini cultures was extracted using alkaline lysis as described elsewhere [107]. Larger-scale plasmid preparations were isolated by silica-based plasmid DNA purification using the NucleoBond^®^ Xtra Midi Kit and Maxi Kit (Macherey Nagel, Düren, Germany), according to the manufacturer’s instructions. Cycle sequencing reactions were prepared using the BigDye Terminator v1.1 Cycler sequencing kit (Applied Biosystems, Foster City, CA, USA) according to the manufacturer’s instructions, and analyzed on a Hitachi 3031xl Genetic Analyzer with Sequence Detection Software version 5.2 (Applied Biosystems, Foster City, CA, USA).

### 4.8. T7 Endonuclease I (T7EI) Assay

Genome editing efficiency at the CRISPR/Cas9 target locus in pools of edited cells was assessed using T7EI assay. A PCR amplicon of the CRISPR/Cas9-targeted sequence was generated using Q5 Hot Start High-Fidelity DNA Polymerase (New England Biolabs, Ipswich, MA, USA), primer pairs listed in Appendix A, and 30 PCR cycles. Amplicons were then purified with the QIAquick PCR Purification Kit (Qiagen, Hilden, Germany) according to the manufacturer’s instructions, denatured at 95 °C for 5 min and reannealed by slow cooling to 35 °C at −0.1 °C/s, for the formation of heteroduplexes, which were then cleaved by incubation with T7EI (New England Biolabs, Ipswich, MA, USA) according to the manufacturer’s instructions. To determine the percentage of genome editing, cleavage products were separated on a 2.5% agarose gel, prestained with RedSafe Nucleic Acid Staining Solution (iNtRON Biotechnology, Inc., Korea), for quantification of band intensities in ImageJ [108] as basis of standard ratio calculations [51].

### 4.9. Tracking of Indels by DEcomposition (TIDE)

For the assessment of genome editing efficiency at the CRISPR/Cas9 target locus in pools of edited cells, the web-based tool TIDE was also utilized [109]. Results of Sanger DNA sequencing of PCR amplicons of CRISPR/Cas9-treated and mock-treated cell samples generated for T7EI assay (2.10) were uploaded along with the gRNA sequence to TIDE for quantification of editing efficacy and identification of the predominant types of mutations (indels) created by editing.

### 4.10. Restriction Fragment Length Polymorphism Analysis of PCR-Amplified Fragments (PCR-RFLP)

PCR-RFLP [110] was used to quantify the percentage of NHEJ-mediated genomic integration of dsODNs or HDR-mediated genomic integration of ssODNs in the 3′ UTR of *BCL11A*. Briefly, PCR products of 300–500 bp containing the DSB-enabled tag-insertion site were digested overnight with HaeIII (for dsODNs, ssODN451TS-2MRSs, ssODN451NTS-2MRSs and Alt-R HDR-2MRSs) and DdeI (for ssODN451TS-1MRS and ssODN451NTS-1MRS) restriction enzymes (New England Biolabs, Ipswich, MA, USA) in CutSmart buffer according to the manufacturer’s instructions. The reactions were separated on a 2% agarose gel, and gel bands were quantified using ImageJ.

### 4.11. Reversed-Phase High-Performance Liquid Chromatography (RP-HPLC) Analysis of Globin Chains

The pellet of 1 × 10^6^ differentiated cells was washed with DPBS and resuspended in 50 μL of HPLC-grade water before two rounds of freezing/thawing. After centrifugation at 16,000× *g* for 10 min at 4 °C, the supernatant was transferred to HPLC vials (Altmann Analytik, Munich, Germany). An LC-20AD chromatographic system (Shimadzu, Kyoto, Kyoto, Japan) and an Aeris Widepore C18 column (Phenomenex, Torrance, CA, USA) were used to separate peptides based on their hydrophobicity and on an increasing linear gradient of acetonitrile with 0.1% trifluoroacetic acid against 0.1% trifluoroacetic acid/0.033% sodium hydroxide for elution from the column, as previously published [111]. An amount of 25–30 μL of protein extract was injected per analysis. Heme and globin chains were eluted from the column at different retention times and detected as absorbance peaks, the area of which was used to determine the relative quantities of globin chains in samples.

### 4.12. Immunoblotting

Consistent with previously published procedures [112], the lysate of 0.5–1 × 10^6^ cells per sample was separated by SDS polyacrylamide gel electrophoresis and transferred onto a Nitrocellulose Parablot NCP membrane (Macherey Nagel Düren, Germany) by wet electroblotting. After blocking, membranes were incubated overnight with primary antibodies (Appendix A), before washing and incubation with corresponding secondary antibodies (Appendix A). Bands were detected using chemiluminescence staining buffer (Lumisensor, GenScript, Piscataway, NJ, USA) and a Biospectrum 810 Imaging System (Thermo Fisher Scientific, Waltham, MA, USA) or Vilber FUSION Solo X (Vilber Lourmat S.A, France). Quantifications were based on mean gray value of bands calculated in ImageJ.

## 5. Conclusions

Taken together, our findings demonstrate the unexploited potential of utilizing the endogenous miRNA machinery for temporal or spatial segregation of endogene expression. By exploring different methods, donors and repair pathways for tagging by hit-and-run editing, our study revealed both limitations of TAMED for therapeutic development based on currently available methodology, and potential suitability of TAMED for the lineage-specific dissection of gene function in research applications.

## Figures and Tables

**Figure 1 ijms-23-01082-f001:**
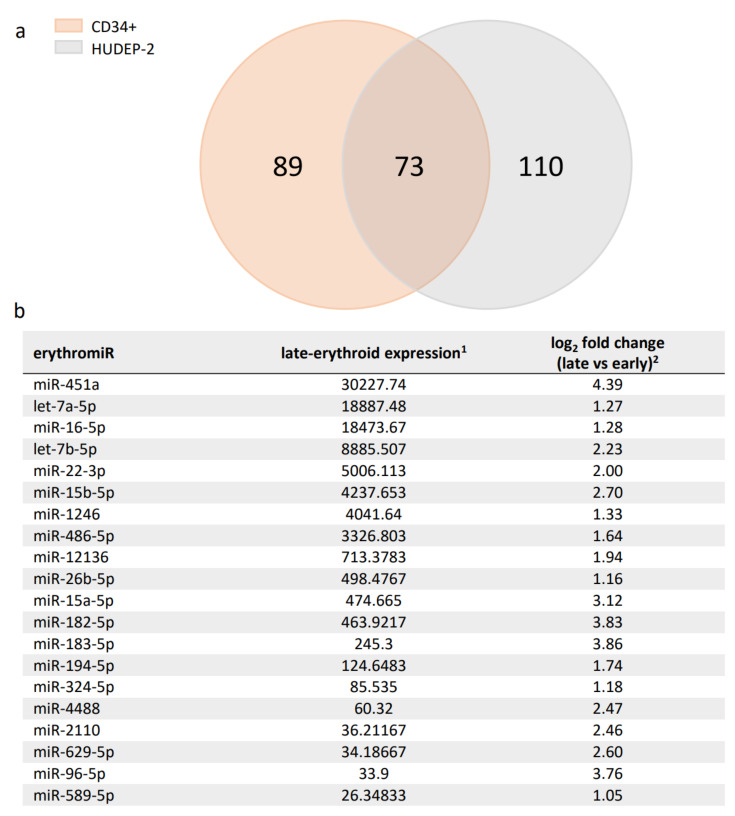
Summary of detected erythromiRs in CD34+ and HUDEP-2 cells. (**a**) DNA Nanoball Small RNA Sequencing and DEGseq differential expression analysis identified 162 and 183 significantly upregulated miRNAs during erythroid differentiation in CD34+ and HUDEP-2 cells, respectively. A Venn diagram of these miRNAs revealed a set of 73 miRNAs common to both cell sources, likely representing key adult-type erythromiRs. (**b**) The top 20 erythromiRs were sorted by late-erythroid expression in CD34+ and HUDEP-2 cells (see Appendix A for all 73 miRNAs). ^1^: mean normalized miRNA counts in late-erythroid samples and ^2^: mean log_2_ fold change of miRNA expression in late- vs. early-erythroid samples, both as calculated by DEGseq [37].

**Figure 2 ijms-23-01082-f002:**
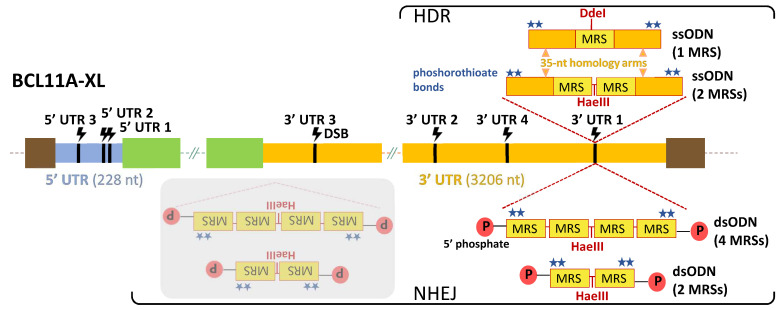
TAMED tools and modes of delivery. Potential integration sites (CRISPR/Cas9-mediated DSB sites) of MRSs are indicated in 5′ UTR (blue) and 3′ UTR (orange) of the BCL11A-XL isoform. Designs and delivery modes of blunt-ended dsODNs (bearing two or four tandem repeats of MRSs for miR-451a) and ssODNs (bearing one or two tandem repeats of MRSs for miR-451a) are schematically illustrated. Representative of all seven UTR sites, potential insertion events are only indicated for 3′ UTR 1, 593 nt upstream from the end of the BCL11A-XL long 3′ UTR (3206 nt). Shown with grey overlay are inverted integration events that will occur as undesirable side products for NHEJ-mediated tagging with blunt-ended dsODNs. DSB: double-strand break, dsODN: double-stranded oligodeoxynucleotide, HDR: homology-directed repair, MRS: miRNA recognition site, NHEJ: non-homologous end joining, P: phosphate group, ssODN: single-stranded oligodeoxynucleotide, UTR: untranslated region (light blue: 5′ UTR, orange: 3′ UTR), green boxes: non-UTR BCL11A-XL sequence elements, brown boxes: flanking genomic DNA elements.

**Figure 3 ijms-23-01082-f003:**
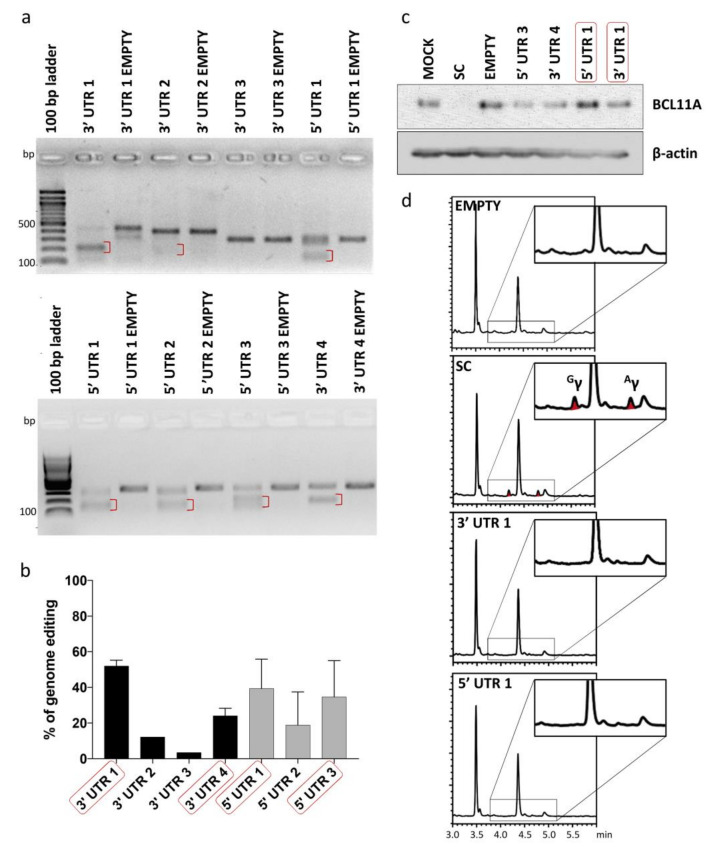
Analysis of genome editing of 5′ and 3′ UTR of BCL11A-XL. (**a**) Representative gel images showing full-length PCR products and T7EI cleavage products (red brackets) of 5′ and 3′ UTR-edited sites. A Cas9-only transduced sample (EMPTY) was also analyzed in parallel to assess specificity of T7EI cleavage. (**b**) Quantification of T7EI assay was based on mean gray value of gel bands in ImageJ, using the 1–(1–(fraction cleaved))^1/2^ formula [51] for 3′ UTR 1 (n = 3), 3′ UTR 2 (n = 1), 3′ UTR 3 (n = 1), 3′ UTR 4 (n = 2), 5′ UTR 1 (n = 3), 5′ UTR 2 (n = 2) and 5′ UTR 3 (n = 2). Error bars show the standard deviation of the sample mean. (**c**) Immunoblot analysis of BCL11A expression in samples edited with shortlisted gRNAs (red boxes in (**b**)) confirmed the neutral effect of editing on gene expression for at least two gRNAs, 3′ UTR 1 and 5′ UTR 1 (**d**) Chromatograms showing the absence of γ-globin in shortlisted samples (red boxes in (**c**)) except for SC, which showed 5.14-fold induction of γ-globin relative to EMPTY (peaks corresponding to γ-globin are indicated as ^G^γ and ^A^γ). EMPTY: Cas9-only-transduced sample, MOCK: mock-transduced cell sample, SC: cell sample transduced with gRNA targeting the start codon of *BCL11A*.

**Figure 4 ijms-23-01082-f004:**
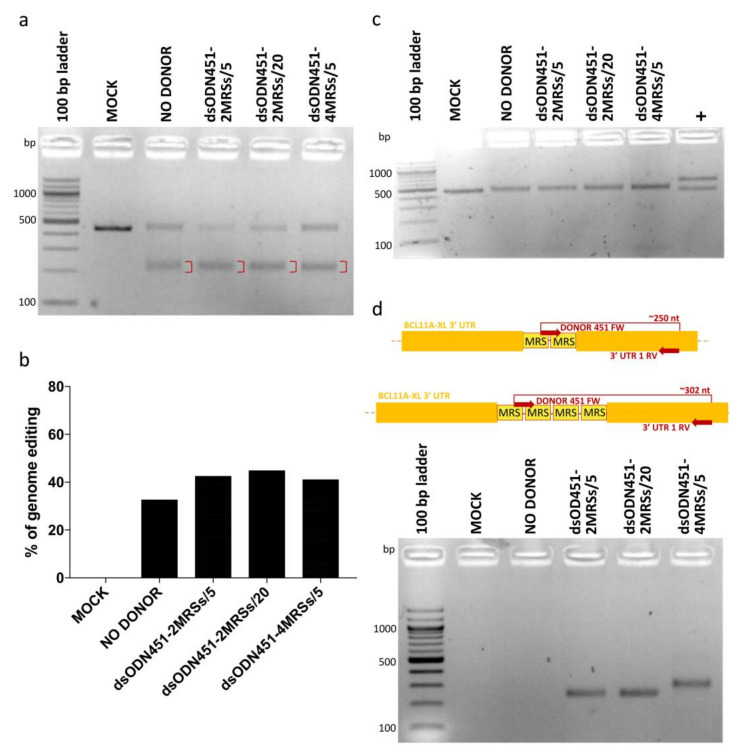
NHEJ-mediated integration of MRSs in HEK293T cells. (**a**) Gel image showing T7EI cleavage products (red brackets) of edited BCL11A-XL 3′ UTR 1 site. (**b**) Quantification of T7EI gel bands showing similar percentages of genome editing across samples (32.6% to 44.8%). (**c**) Apparent absence of PCR-RFLP HaeIII cleavage for test samples, indicative of dsODN integration being below the detection threshold. (**d**) Top: schematic illustration of dsODN-specific amplification assay at the 3′ UTR 1 site; bottom: amplicons of ~250 and ~302 bp revealing the presence of 2 MRSs and 4 MRSs in cells, respectively. Of note, undesirable inverted integration of MRSs into 3′ UTR 1 (see greyed-out part of Figure 2) is not detected by this assay. MOCK: mock-nucleofected cell sample, NO DONOR: cell sample nucleofected only with RNPs, MRS: miRNA recognition site, dsODN451-2MRSs/*: cell samples nucleofected with RNPs and dsODN bearing two miR451a MRSs at the indicated picomole quantity, dsODN451-4MRSs/5: cell sample nucleofected with RNPs and 5 pmole dsODN bearing four miR451a MRSs, +: a 994-bp PCR product giving cleavage products of 572 and 422 bp after digestion with HaeIII.

**Figure 5 ijms-23-01082-f005:**
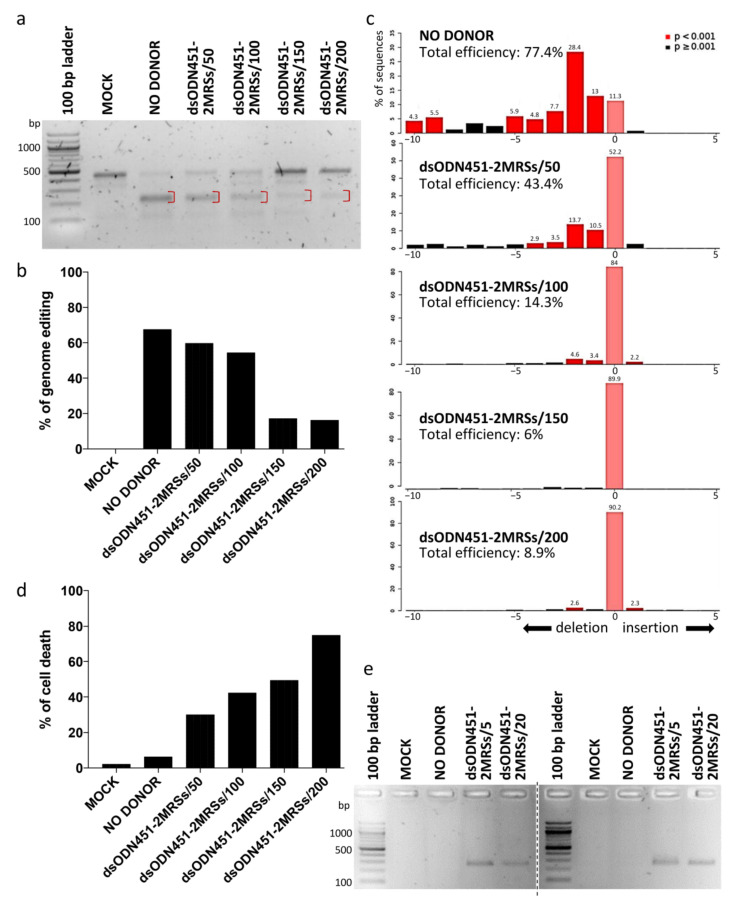
NHEJ-mediated integration of MRSs in HUDEP-2 and CD34+ cells. (**a**) Gel image showing T7EI cleavage products (red brackets) of edited BCL11A-XL 3′ UTR 1 site in HUDEP-2 cells. (**b**) Quantification of T7EI gel bands from (**a**), showing dsODN-concentration-correlated reduction in genome editing across samples. (**c**) Quantification of indel formation (genome editing efficiency) by TIDE for comparison with T7E1 assay results from (**a**). While showing slightly higher total efficiency for NO DONOR (77.4%) and lower efficiencies for all other samples (6–43.4%) than the T7E1 assay, TIDE gave the same overall trend of diminished total editing efficiency with increasing amount of donor. TIDE additionally demonstrated that the selected gRNA 3′ UTR 1 induced mainly deletions, with a 2-nt deletion as the most frequent event in the absence (28.4%) or presence (<2–13.7%) of donor DNA. (**d**) Dose-dependent dsODN-mediated cytotoxicity in HUDEP-2 cells, as measured by trypan blue assay 72 h after nucleofection. (**e**) Indicative of dsODN integration, a dsODN-specific amplification assay at the 3′ UTR 1 site gave ~250 bp products in both HUDEP-2 (left gel) and primary CD34+ cells (right gel). Separate gels are indicated by a dashed line. MOCK: mock-nucleofected cell sample, NO DONOR: cell sample nucleofected only with RNPs, MRS: miRNA recognition site, dsODN451-2MRSs/*: cell samples nucleofected with RNPs and dsODN bearing two miR451a MRSs at the indicated picomole quantity.

**Figure 6 ijms-23-01082-f006:**
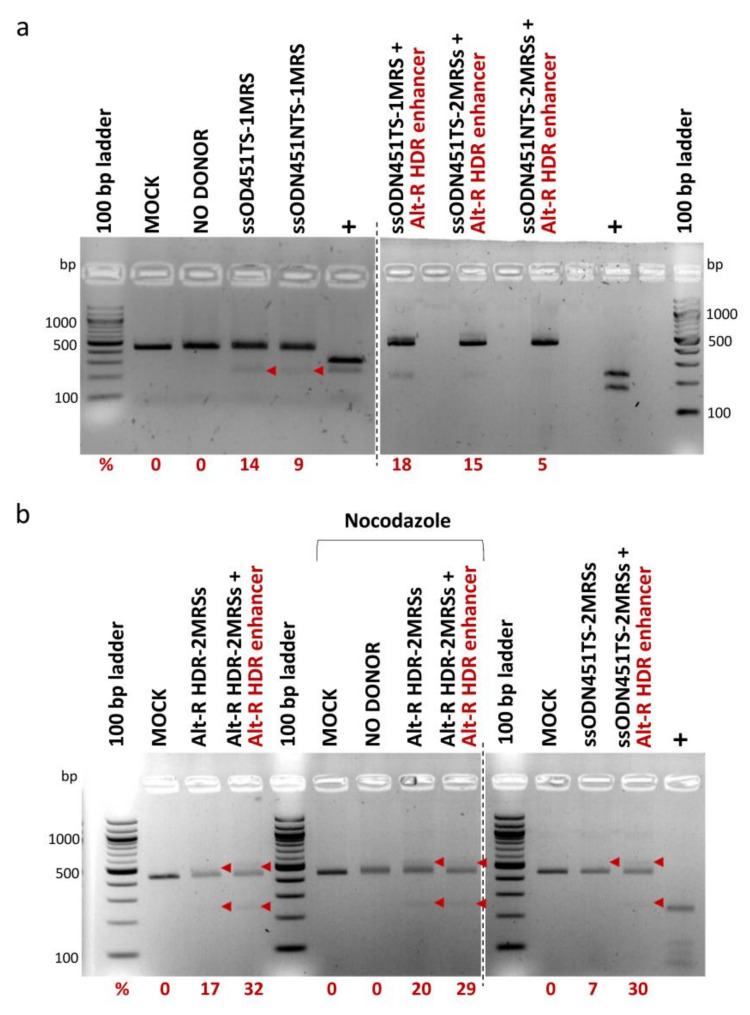
HDR-mediated integration of MRSs in HUDEP-2 cells. PCR-RFLP analysis of cells nucleofected with RNPs and ss DNA oligos was performed for the assessment of integration efficiency. Bands of 436 bp represent unmodified target sites, larger bands insertions with incomplete digestion by DdeI/HaeIII, and smaller bands cleavage products of insertions. All bands corresponding to insertions are indicated by arrowheads (band sizes of 554 bp and 277 bp for donors with 2 MRSs, and of 528 bp and 264 bp for donors with 1 MRS). Corresponding rates of DdeI/HaeIII cleavage in PCR-RFLP (%) (after subtraction of background average cleavage rates of control samples) are reported below the gels. (**a**) Analysis of cells nucleofected with RNPs and Ultramer DNA Oligos. (**b**) Analysis of cells nucleofected with Alt-R RNPs and Alt-R HDR-2MRSs DNA Oligo (left gel), shown in comparison with analysis of cells nucleofected with standard Cas9 RNPs and Ultramer ssODN451TS_2MRSs DNA Oligo (right gel). Separate gels are indicated by dashed lines. MOCK: mock-nucleofected sample, NO DONOR: cell sample nucleofected only with RNPs, MRS: miRNA recognition site, NTS: non-target strand, TS: target strand, +: PCR products bearing restriction sites for DdeI/HaeIII used as positive controls. See Section 4.2 and Appendix A for ss DNA oligo naming.

**Figure 7 ijms-23-01082-f007:**
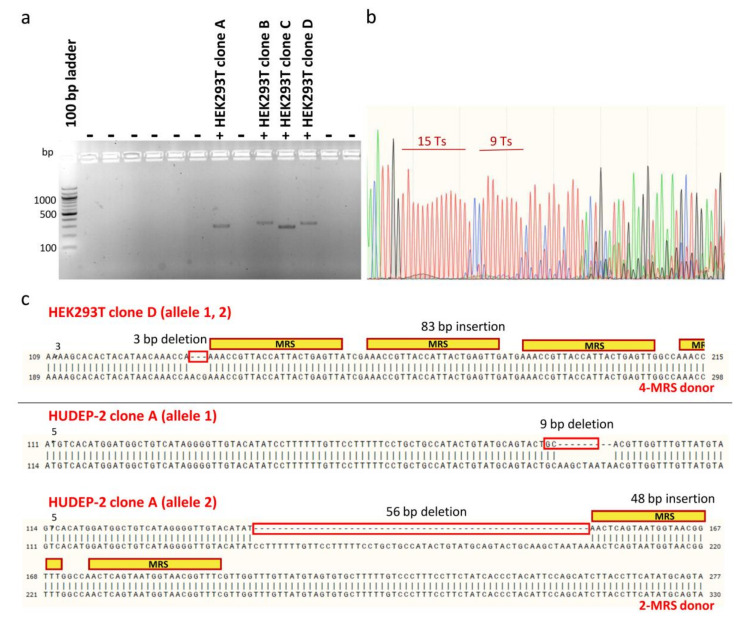
Analysis of monoclonal cell populations at the DNA level. (**a**) Exemplary gel showing initial screening for positive HEK293T clones (clones with MRSs in the correct orientation), based on dsODN-specific amplification. Different sizes of PCR products correspond to integrations of different numbers of MRSs. (**b**) Sequencing chromatogram of the 3′ UTR 1 region for wild-type CONTROL HUDEP-2 clone. Representative of other sequencing data for 3′ UTR 1 clones, sequence trace data become mixed after successive long mononucleotide (T) runs. (**c**) Examples of imprecise insertion events for HEK293T clone D and HUDEP-2 clone A, representative of imprecisions for all detected NHEJ-based dsODN insertion events in both HEK293T and HUDEP-2 cells. For homozygous HEK293T clone D, alignment of its sequence trace with the predicted precise insertion of donor DNA revealed a 3-bp chromosomal deletion and insertion of at least three intact MRSs. For compound heterozygote HUDEP-2 clone A, corresponding alignments revealed a 9-bp chromosomal deletion in one allele and a 56-bp chromosomal deletion and a 48-bp insertion of intact MRSs in the other allele. Inadvertent deletions are indicated by red boxes.

## Data Availability

Not applicable.

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
