# Peer review of "CRISPR Editing Enables Consequential Tag-Activated MicroRNA-Mediated Endogene Deactivation"

_ijms, 2022, doi:10.3390/ijms23031082_

Round 1

Reviewer 1 Report

The article entitled "CRISPR editing enables consequential tag-activated microRNA-mediated endogene deactivation" provides a nice overview of novel tag-activated microRNA (miRNA)-mediated endogene deactivation (TAMED) potential use in molecular therapies and biomedical application. The authors have used CRISPR/CAS9 mediated DSB and integration of miRs using either NHEJ or HDR pathway in the UTRs of BCl11A gene and studied its expression effect. The article can be consider for publication, however I have one major comment which needs suitable explanation from the authors. Below are the comments:

Major comments:

1) Figure 8: It is unclear that how many biological replicates are used in the experiments. Also, authors used two different cell lines (HEK293T and HUDP-2) in their study. Both of these cell line have shown mixed effect of gene suppression. Without any biological replicates it will be hard to understand/determine the statistical significance of results obtained.

Minor Comments:

1) Page 3, Line 117 HEK293T is mentioned to times. Once can be removed.

2. Page 8, Figure 1: Need to annotate different panels with (a) and (b) in figure itself. It is in the legend but not present in the figure.

3) Page 20, Line 630. " on the one hand". Consider replacing this with "First" instead

4) Page 20, Line 631 and Line 634-637. Simplify and rephrase these sentences. It is hard to understand.

5) Page 20, Line 639 "moreover" this is redundant here.

6) Page 20, Line 648-  replace "5" with "5'"

7) Page 20 Line 681-682 and Page 21 Line 683-685. Simplify these statement.

8) Page 21, Line 688 "Achieved" This is not required here.

Author Response

The article entitled "CRISPR editing enables consequential tag-activated microRNA-mediated endogene deactivation" provides a nice overview of novel tag-activated microRNA (miRNA)-mediated endogene deactivation (TAMED) potential use in molecular therapies and biomedical application. The authors have used CRISPR/CAS9 mediated DSB and integration of miRs using either NHEJ or HDR pathway in the UTRs of BCl11A gene and studied its expression effect. The article can be consider for publication, however I have one major comment which needs suitable explanation from the authors. Below are the comments:

[Reply] We thank the Reviewer for this fair and overall positive assessment.

Major comments:

1) Figure 8: It is unclear that how many biological replicates are used in the experiments. Also, authors used two different cell lines (HEK293T and HUDP-2) in their study. Both of these cell line have shown mixed effect of gene suppression. Without any biological replicates it will be hard to understand/determine the statistical significance of results obtained.

[Reply] For each of the 4 HEK293T clones and 10 HUDEP-2 clones shown in Figure 8, a single experimental replicate is shown. While each of the clones may be considered a biological replicate, there is substantial variation between clones, and we appreciate that evidence of reproducibility of knockdown for BCL11A and of induction for γ-globin for each clone would be desirable, and that in particular the dosage-dependent reduction of BCL11A in response to miR-451a-mimic application, albeit consistent within itself, can only be indicative and will be authoritative only with experimental replicates and corresponding statistics. We are therefore faced with a dilemma.      
On the one hand and beyond this point rightly being made by the Reviewer, the data we now have in hand give orders of magnitude better endogene knockdown in bulk and across clones than has been achieved before for miRNA target sequence insertion, and it would therefore after all be warranted to invest in replicate analyses in order to give greater credence to our findings. On the other hand, there is redundancy in our clonal analyses, and our efficiency and toxicity results for the original GUIDE-Seq oligonucleotide perfectly aligned with published data, indicating that our general setup is sound. Owing to our focus on therapeutic applications and once we became convinced, after long delays over toxicity, efficacy and clonal analyses, that TAMED would be unsuitable for therapy development with current donor chemistry for the target in hand, it therefore seemed logical to conclude our analyses of collected biomaterials swiftly (i) in order to avoid further delays for publication in a fast-moving field and (ii) as a meaningful pointer for others, because we wish corresponding data had been available when we originally set out to develop TAMED.     
To our chagrin, functional analyses in particular for HUDEP-2 clones take several weeks to reach evaluable cell numbers, so that even replicate applications of an miR-451a mimic as shown in Figure 8b would take between two and three months to perform. To stay within the resubmission timeframe of seven days, we have now merely added text making single replicates for clones explicit and emphasizing the indicative nature of our data. Naturally and with all the time invested in the data already, we do not want this point to be an impediment to publishing our findings. If the current version of the article is not acceptable, we could alternatively with an additional week of time for resubmission, reformat the paper as a Short Communcation by shortening the main text and moving a substantial part of the data to the supplementary section, hoping that this would find approval.

Minor Comments:

1) Page 3, Line 117 HEK293T is mentioned to times. Once can be removed.

[Reply] Done, also in line with a request by the journal to give the cell origin of third-party cell lines used.

  1. Page 8, Figure 1: Need to annotate different panels with (a) and (b) in figure itself. It is in the legend but not present in the figure.

[Reply] Done; thank you for the vigilance.

3) Page 20, Line 630. " on the one hand". Consider replacing this with "First" instead

[Reply] Prompted by this comment, we now use “As the first limiting effect” and “As the second limiting effect” in order to separate the paragraph and guide the reader better.

4) Page 20, Line 631 and Line 634-637. Simplify and rephrase these sentences. It is hard to understand.

[Reply] Done. The section in question (in a paragraph that we hope we have overall made more accessible) is now as follows:

“Based on our data, utility of TAMED is thus currently limited by two independent ef-fects. As the first limiting effect, NHEJ-mediated integration of dsODNs by CRISPR/Cas9 is nondirectional and imprecise. Nondirectionality for Cas9 brings about that approximately half of all integration events may be reversed and therefore unproductive for TAMED. Imprecision leads to random on-target indel events of flanking gDNA and of sequence tags, in line with previous studies using CRISPR/Cas9 in knock-in experiments [20,99]. Such ubiquity of indels at CRISPR/Cas9 on-target sites may be caused by blunt-ended DSBs. By contrast, the typically staggered cuts of zinc-finger nucleases produce complementary acceptor ends, which facilitate precise donor sequence insertion of dsODNs with microhomology arms in knock-in experiments [100].”

5) Page 20, Line 639 "moreover" this is redundant here.

[Reply] We have replaced “moreover” with “additionally,” which more clearly conveys the idea that truncation of inserts and extended deletion of the UTR are two separate undesirable effects: “This phenomenon effectively reduces the availability of MRSs for interaction with miRNAs by their partial deletion in truncated insert sequences and might additionally interfere with the functionality of the UTR by extended deletion of DSB-flanking genomic DNA.”

6) Page 20, Line 648-  replace "5" with "5'"

[Reply] Done.

7) Page 20 Line 681-682 and Page 21 Line 683-685. Simplify these statement.

[Reply] Done. We have altogether restructured the section containing the lines in question, in order to make the text more accessible. The text now reads as follows:

“Another issue of import for any selected miRNA is whether its isolated exploitation for TAMED will be able to evoke clinically relevant suppression of a target gene given the “multiple-to-multiple” nature of small RNA interactions with targets, in accordance with our own transcriptomic data [37]. Importantly, for the highly expressed miR-451a, our clonal TAMED data show consequential suppression of BCL11A with ensuing γ-globin increases, and as a natural genetic phenomenon, 3’ UTR mutations that abrogate natural or create illegitimate MRSs elsewhere have been shown to cause disease [109,110]. However, these might be exceptional observations, as suggested by several studies and because knockouts of individual miRNAs in erythroid cells have so far failed to produce significant phenotypic changes related to γ-globin [10,111–113]. Therefore, insertion of different concatenated MRSs for two or three different erythromiRs may reach higher gene suppression than could be achieved employing multiple MRSs for mi-R451a alone, even if the corresponding miRNAs are expressed at moderate levels [114]. This strategy would at the same time reduce the risk of saturating the function of any one cognate miRNA [111].”

8) Page 21, Line 688 "Achieved" This is not required here.

[Reply] Done. In line with Minor Comment 7 above, we have changed the clause in question for clarity: “…insertion of different concatenated MRSs for two or three different erythromiRs may reach higher gene suppression than could be achieved employing mi-R451a alone…”

Reviewer 2 Report

CRISPR editing enables consequential tag-activated microRNA-mediated endogene deactivation

Panayiota L. et al describe the deactivation of BCL11A through the insertion of one/several MRSs for miR-451a at its 5´and 3´UTR by NHEJ and HDR. They discuss then, on base of their results, the implications that TAMED may have on research and gene therapy.

The manuscript is of a high quality. It is written in a well-designed, concise, and clear way, which makes the layout very easy to follow. I have just two minor corrections and a question.

Line 117: “Embryonic kidney 293T cells (HEK293T cells) HEK293T cells…” (repeated)

In figure 3d) does the second chromatogram correspond to SC? If so, might you write SC at it to make the figure clearer?

Have you assessed any possible off-target indels or large truncations of gDNA in the erythroid cells after your HDR-TAMED assay?

Author Response

Panayiota L. et al describe the deactivation of BCL11A through the insertion of one/several MRSs for miR-451a at its 5´and 3´UTR by NHEJ and HDR. They discuss then, on base of their results, the implications that TAMED may have on research and gene therapy.

The manuscript is of a high quality. It is written in a well-designed, concise, and clear way, which makes the layout very easy to follow. I have just two minor corrections and a question.

[Reply] We thank the Reviewer and are delighted about this positive assessment.

1) Line 117: “Embryonic kidney 293T cells (HEK293T cells) HEK293T cells…” (repeated)

[Reply] Done, also in line with a request by the journal to give the cell origin of third-party cell lines used.

2) In figure 3d) does the second chromatogram correspond to SC? If so, might you write SC at it to make the figure clearer?

[Reply] Done. A mistake in image layers had obscured the chromatogram label; thank you for spotting this oversight.

Have you assessed any possible off-target indels or large truncations of gDNA in the erythroid cells after your HDR-TAMED assay?

[Reply] With efficiency as our primary concern, our fidelity- and safety-related analyses were limited to (i) the in silico analyses and choice of highest-scoring gRNAs in the initial design of UTR-targetting CRISPR/Cas nucleases and (ii) the experimental characterization of on-target indels for our clonal analyses, both as detailed in the manuscript. Combination of 3’ UTR 1 gRNA with the HDR donor designs analysed would have been unsuitable for therapeutic application for reasons of knockdown efficiency and toxicity, so that experimental evaluation of the specific corresponding off-target and recombination events was not a priority. However, we agree with the Reviewer that this point should have been made in our Discussion of TAMED, in particular because for any application going forward for therapy, the suggested analyses would be obligatory. We have now added the utility of additional fidelity and safety analyses to the Discussion.

Round 2

Reviewer 1 Report

In the revised article entitled "CRISPR editing enables consequential tag-activated microRNA-mediated endogene deactivation" authors have adequately addressed my all concerns however I have a following suggestion for my major concern I raised earlier for the biological replicates for Fig. 8

1) If the authors thinks generating the replicate data for HUDP2 cells will not be feasible within the give time frame than the results for HUDEP2 cells (Fig. 8C) can be removed from main manuscript and present it as the supplementary data.

2) Also rather using single replicates I think it would be better to state it as "sample (n=1) per clone was used" to avoid any confusion.

After taking care of these issues the manuscript should be ready for publication.

Author Response

In the revised article entitled "CRISPR editing enables consequential tag-activated microRNA-mediated endogene deactivation" authors have adequately addressed my all concerns however I have a following suggestion for my major concern I raised earlier for the biological replicates for Fig. 8

1) If the authors thinks generating the replicate data for HUDP2 cells will not be feasible within the give time frame than the results for HUDEP2 cells (Fig. 8C) can be removed from main manuscript and present it as the supplementary data.

[Reply] Done. We thank the Reviewer for the suggested solution. For consistency and with clonal data for HEK293T and HUDEP-2 cells already represented (for sequence analyses) in Figure 7, we have now moved all of Figure 8 to the supplementary section as Supplementary Figure S2, renumbering Supplementary Materials accordingly.

2) Also rather using single replicates I think it would be better to state it as "sample (n=1) per clone was used" to avoid any confusion.

After taking care of these issues the manuscript should be ready for publication.

[Reply] Done.